# Polypharmacy and associated factors in South Korean elderly patients with dementia: An analysis using National Health Insurance claims data

**Hea-Lim Kim[1], Hye-Jae Lee**[ID][2]*

**1** Department of Public Health Sciences, Graduate School of Public Health, Seoul National University, Seoul, Republic of Korea, **2** Department of Environmental Health, Korea National Open University, Seoul, Republic of Korea

\* sangtu2@gmail.com

## Abstract

**Data Availability Statement:** This study used the NHIS-Senior cohort database curated by the National Health Insurance Service (NHIS) of Korea. This data cannot be shared publicly due to NHIS

### Background

Dementia is accompanied by several symptoms, including cognitive function decline, as well as behavioral and psychological symptoms. Elderly patients with dementia often experience polypharmacy, the concurrent use of multiple medications, due to chronic comorbidities. However, research on polypharmacy in patients with dementia is limited. This study aimed to characterize polypharmacy and associated factors among elderly patients with dementia in South Korea, and compare the characteristics of patients with and without dementia patients.

### Methods

From the National Health Insurance Service (NHIS)-Senior cohort database, we extracted data on patients aged≥60 years who received outpatient treatment in 2019. Polypharmacy was defined as the concurrent use of five or more different oral medications for ≥90 days; excessive polypharmacy referred to the concurrent use of ten or more different oral medications for ≥90 days. We compared the prevalence of polypharmacy between patients with and without and identified the associated factors using a logistic regression model.

### Results

About 70.3% and 23.7% of patients with dementia exhibited polypharmacy and excessive polypharmacy, respectively. After adjusting for conditions such as age and Charlson's comorbidity index, the likelihood of polypharmacy and excessive polypharmacy significantly increased over time after the diagnosis of dementia. Additionally, under the same conditions, Medical Aid beneficiaries with dementia were more likely to experience polypharmacy and excessive polypharmacy compared to patients with dementia covered by National Health Insurance (NHI).

regulations. We obtained approval for use of the data after review by the Institutional Review Board of the affiliated institution and a separate review by the NHIS, and paid an access fee. We were able to access the data for the approved period through the remote service operated by the NHIS. Those who are eligible for use of the National Health Information Data, as stipulated by the NHIS, can access the NHIS-Senior cohort database following the same procedure mentioned here. Applications can be made with the National Health Insurance Sharing Service (https://nhiss.nhis.or.kr/bd/ab/bdaba021eng.do).

**Funding:** HJL and HLK was supported by Basic Science Research Program through the National Research Foundation of Korea (NRF) (grant No.2021R1F1A1062230). The funders have played no role in study design, data collection and analysis, decision to publish, or preparation of the manuscript.

**Competing interests:** The authors have declared that no competing interests exist.

## Conclusion

This study reports the latest evidence on the status and risk factors of polypharmacy in elderly patients with dementia. We proposed that careful monitoring and management are required for patients at high risk for polypharmacy.

## Introduction

Dementia refers to a decline in cognitive functions including memory, language, and problem-solving, resulting from various causes, which significantly impairs quality of life [1]. While non-pharmacological interventions, such as cognitive and physical exercises, may improve cognitive function and delay the progression of disease, pharmacological treatment remains the primary clinical approach [2, 3]. Antipsychotics are often prescribed to treat the behavioral and psychological symptoms of dementia (BPSD) [4–6]. Moreover, elderly individuals with dementia often have comorbid chronic diseases [7]. Given that multiple pharmacological treatments may be prescribed to manage existing chronic diseases, often managed under fragmented healthcare systems with disease-specific treatment guidelines, patients with dementia are prone to polypharmacy [8–11].

Polypharmacy is commonly defined as the daily use of more than five medications, though criteria differ regarding the duration of medication exposure and inclusion of over-the-counter or traditional and complementary medicines [12, 13]. Given the potential adverse impacts on health and the efficacy of concurrently used treatments, polypharmacy is recognized as an important public health issue [12, 14, 15].

Polypharmacy in patients with dementia requires caution in several aspects, especially since its long-term effects are largely unknown [10, 11]. Given that dementia predominantly affects the elder population, changes in pharmacokinetics and pharmacodynamics may occur [16], yet most medications lack clinical trial evidence specifically for elderly patients [17]. Some associations have been reported between polypharmacy and function decline, cognitive impairment, and falls in elderly patients [18]. Cognitive function decline and memory loss (common symptoms of dementia) can further interfere with the identification of side effects and symptoms related to co-prescribed medications [19, 20]. Hence, despite the importance of addressing polypharmacy in patients with dementia, studies on this issue remain limited. Previous research has primarily focused on nursing home settings or potentially inappropriate medications, while studies on polypharmacy and associated factors in outpatient patients with dementia are relatively scarce [8, 20–23].

With the aging global population, the number of individuals diagnosed with dementia increased by 117% worldwide between 1990 and 2016 [24], and is expected to continue increasing in future [25]. Consequently, the management of patients with dementia has become a crucial clinical service [26–28]. In South Korea, one of the fastest aging countries, 10.3% (924,870 people) of the population aged≥65 years suffered from dementia in 2022, which is 1.7-times higher than that in 2012. The prevalence of dementia in South Korea was predicted to increase by another 1.6-fold by 2032 [29, 30]. Consequently, dementia has gained national research interest in South Korea, especially regarding its pharmacological treatment [31–36]. Nevertheless, no study has addressed the characteristics of polypharmacy in patients with dementia. The study of polypharmacy in patients with dementia could provide valuable information not only for clinical practitioners and policy makers in Korea, where the population is aging rapidly, but also in other countries with slower population aging rates. In

addition, a recent systematic literature review found that the prevalence of polypharmacy in patients with dementia varied considerably between regions [23]. This may reflect the influence of region-specific healthcare systems and the different settings of each study. The results of the study using representative data from Korea are expected to add additional evidence on polypharmacy in dementia patients, especially to the lack of evidence in Asian countries. To bridging this gap in knowledge, we aimed to determine the status and associated factors of polypharmacy among outpatients with dementia in 2019, using data sourced from the National Health Insurance Service (NHIS)-Senior cohort database. To elucidate the characteristics, prevalence, and risk factors of polypharmacy in dementia, we compared the data between patients with and without dementia.

## Materials and methods

### Data source and ethics

Data were sourced from the NHIS-Senior cohort database (2002–2019). The sample dataset accounted for 8% of the South Korean population aged 60–80 years as of 2008. Follow-up in this cohort was performed until 2019. Each year, about 8% of new individuals turned 60 were added to the cohort, and these were also followed until 2019.

The database includes comprehensive information on the utilization of healthcare services covered by National Health Insurance (NHI) and Medical Aid in South Korea, as well as demographic and socioeconomic information, medical and prescription records generated from healthcare visits, and long-term care utilization by the elderly [37]. NHI covers ~97% of the population, and Medical Aid for low-income individuals covers the remaining ~3%. Based on a cross-sectional study design, we used NHIS-Senior data for 2019—the most recent dataset. The data were de-identified of any personal information by the NHIS and provided remotely so that it could not be exported. For the analysis of this study, NHIS-Senior data were accessed from December 16, 2022 to June 19, 2023. Ethical clearance for the study was waived by the Institutional Review Board (IRB) of Woosuk University (October 20, 2021, at the beginning of the study; IRB no. ABN01-202305-02-V1). Due to a change in the corresponding author's affiliation, ethical clearance exemption was again granted by the IRB of Korea National Open University (May 17, 2023; IRB no. ABN01-202305-02-V1).

### Study population

This study included patients who used outpatient services in 2019 and had one or more oral medication prescription. Those who died before March 31, 2019, for whom the period corresponding to the polypharmacy definition criteria (see the **Variables** section) could not be observed, were excluded from the study. Based on previous definitions in similar studies [33, 34, 38], patients with dementia were identified as those who received at least one treatment with the primary or secondary disease codes F00, F01, F02, F03, G30, G31.82 (Korean Standard Classification of Diseases and Causes of Death-7; KCD-7; S1 Table), or those who were prescribed dementia medication (donepezil, galantamine, rivastigmine, and memantine) at least once in 2019. These criteria included not only patients with Alzheimer's disease, vascular dementia, and dementia in other diseases, but also patients taking dementia medication without the aforementioned disease code.

### Variables

Polypharmacy was defined as the concurrent use of five or more oral medications for $\geq 90$ days; different medications were defined based on the World Health Organization Anatomical

Therapeutic Chemical (WHO ATC) fourth-level classification. The definitions referred to the OECD methods of Health Care Quality and Outcomes Indicators [39]. The ATC code classifies medications into five hierarchical levels; the fourth level is defined according to the chemical, pharmacological, or therapeutic subgroups [40]. Excessive polypharmacy was defined as the simultaneous use of 10 or more different (based on the ATC fourth-level classification) oral medications for ≥90 days. All medications prescribed in 2019 were included; therefore, cases with prescription end dates (i.e., date the medication was prescribed + number of days prescribed) extending beyond 2019 were also included.

Age was categorized into 5-year intervals based on an individual's age in 2019. Disability was classified into "severe disability" (Grades 1 to 3), "mild disability" (Grades 4 to 6)—based on national criteria [37]—and "no disability." The disability grade is determined according to the criteria set for each type of disability by the Ministry of Health and Welfare notice when registering a disability [41]. The residential region was classified into five regions, i.e., Seoul-Metro (Seoul–Gyeonggi-do–Incheon), Chungcheong (Daejeon–Sejong–Chungcheongbuk-do–Chungcheongnam-do), Honam (Gwangju–Jeollanam-do–Jeollabuk-do), Gyeongsang (Busan–Daegu–Ulsan–Gyeongsangbuk-do–Gyeongsangnam-do), and Gangwon-Jeju (Gangwon-do–Jeju), based on location and administrative district. Most of the study population had no missing values, but a few had missing residential data. For these individuals, we imputed missing values based on the individual's residential information for the previous year/-s. NHI patient income levels were divided into quintiles based on insurance contribution (the fifth quintile corresponds to the highest contribution), while Medical Aid beneficiaries, representing the lowest income group, were treated as a separate category. The type of long-term care benefits was classified, based on the earliest long-term care benefit records in 2019, as "institutional care" services provided upon admission to a nursing or community living home, "home care" services provided by a visiting caregiver or nurse, and "none" for those who did not receive long-term care benefits. Cases where there were both institutional and home care benefits for the same payment date were defined based on the long-term care benefit record for the following date.

The duration of dementia was estimated from the first date of treatment with the dementia diagnosis code to January 31, 2019. A duration of <1 year was given in cases where dementia medication was prescribed in 2019 but no dementia diagnosis code was provided before 2019. Charlson's comorbidity index (CCI) to identify comorbid conditions using the weights provided by Quan et al. [42]. Although it varies slightly different from that in our study, we used the definition of dementia assumed by the CCI algorithm when calculating CCI [43]. All diseases recorded in the 2019 medical records were used to calculate the CCI, including all diseases recorded after the primary and secondary diseases for each visit, except for any ruled-out diagnoses. Considering the weight for dementia (i.e., 2) and the CCI distribution of the study population, CCI was categorized as "≤2," "3–4," "≥5."

## Analysis

We divided the study population into patient groups with and without dementia for the primary analysis, and evaluated the prevalence of polypharmacy and distribution of patient characteristics in each group. The chi-square test was used to assess the differences between the groups. We evaluated the distribution of patient characteristics in the polypharmacy and excessive polypharmacy groups. A chi-square test was performed to compare patients with and without polypharmacy, as well as patients with and without excessive polypharmacy. Multivariate logistic regression was conducted to identify the factors associated with polypharmacy and excessive polypharmacy. For comparison, the second and third analyses were also

conducted in patients without dementia; however, considering the difficulty in defining the duration of dementia in these patients, the analyses excluded this variable, and again, for comparison, the third analysis was also conducted excluding this variable in patients with dementia. Multicollinearity between covariates in each model was assessed using the generalized variance inflation factor calculated using the car package [44] in R software version 4.3.0 (The R Foundation, Vienna, Austria).

To better understand polypharmacy patients with dementia, we performed additional analyses on the distribution of some related comorbidities and the associated medication composition. The additional comorbidities were hypertension, depression, and mental health-related diseases other than dementia, based on definition in previous studies [45] and the available data. Regarding medication composition, we investigated the prevalence of dementia medications and medications associated with BPSD. Medications were identified based on the ATC code; the ATC codes of medications related to BPSD were based on previous studies [46, 47]. The medications investigated were as follows: N06D (anti-dementia drugs), N05A (antipsychotics), N05B (anxiolytics), N05C (hypnotics and sedatives), N03A (antiepileptics), N06A (antidepressants), N02 (analgesics). We examined respectively the presence of these medications in the medication combinations of days with 5 or more concurrent medications for polypharmacy patients and 10 or more concurrent medications for excessive polypharmacy patients. All statistical analyses were performed using SAS Enterprise Guide 8.3 (SAS Institute, Cary, NC, USA), unless otherwise stated.

## Results

In the NHIS-Senior cohort, 869,631 outpatients were prescribed oral medications in 2019. Among them, 1,954 patients died before March 31, 2019, thus a total of 867,677 patients were included in the final analysis. Among them, 57,346 and 810,331 were defined as patients with and without dementia, respectively (Fig 1).

The patients with and without dementia showed significant differences in several characteristics (Table 1). Specifically, the dementia group was, on average, 10 years older than the non-dementia group. Additionally, 10.0% of the dementia group fell into the 60s age range, compared to 56.6% of the non-dementia group. The proportion of individuals with disabilities in the dementia group was 27.8%, which was more than twice that in the non-dementia group (12.7%). Long-term care benefits were provided to 48.3% of patients with dementia, but only 2.5% of patients without dementia. Nearly 33.0% and 78.1% of patients with and without dementia had a CCI score ≤2, respectively. (The detailed distribution of comorbidities included in the CCI is presented in S2 Table). Polypharmacy was recorded in 70.3% and 37.4% of patients with and without dementia, respectively, whereas excessive polypharmacy was recorded in 23.7% and 6.8% of patients with and without dementia, respectively (Table 1).

There was no significant difference in the distribution of sexes between patients with and without polypharmacy among dementia group (Table 2). However, the proportion of males was significantly higher in patients with excessive polypharmacy than in those without excessive polypharmacy (34.0% compared to 31.4%). Mild disability was observed among 23.0% of patients with excessive polypharmacy, which was significantly higher than that among patients without excessive polypharmacy (17.3%). Medical Aid beneficiaries were 20.8% of patients with excessive polypharmacy (compared to 12.9% of patients without excessive polypharmacy) and 15.8% of patients with polypharmacy (compared to 12.2% of patients without polypharmacy). In terms of long-term care benefits, home care benefits were provided to 36.5% and 32.2% of patients with and without excessive polypharmacy, respectively. Patients with polypharmacy and excessive polypharmacy showed a high CCI. A CCI score≥5 was observed in

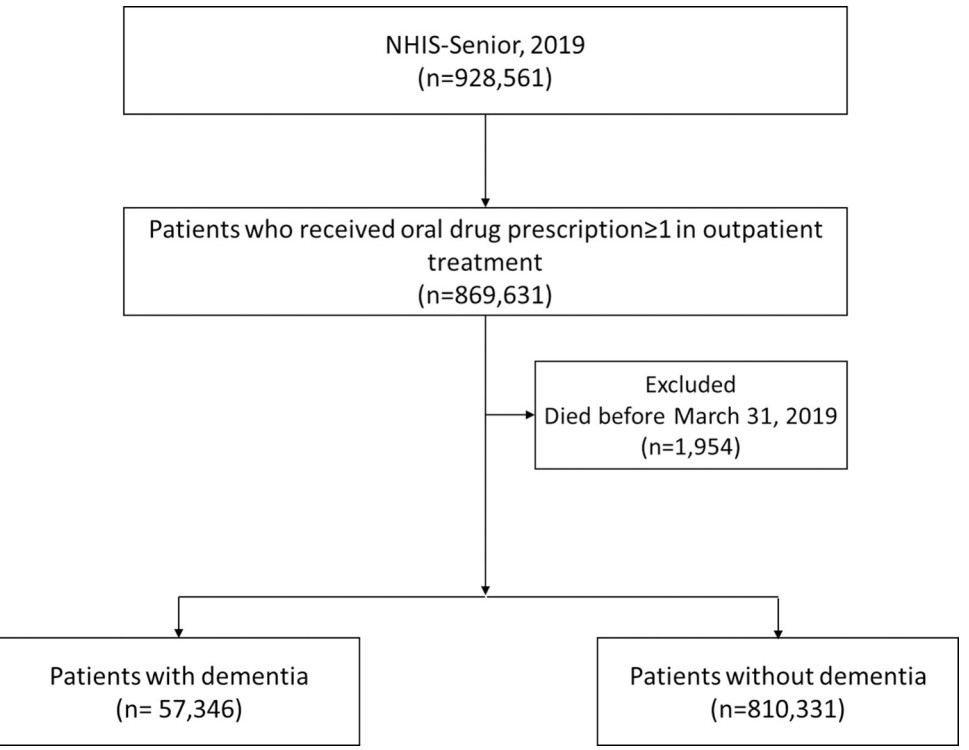

**Fig 1. Selection of study population.**

48.5% and 25.8% of patients with and without excessive polypharmacy, respectively (Table 2). The detailed distribution of comorbidities included in the CCI is shown in S3 Table. The prevalence of congestive heart failure, chronic pulmonary disease, mild liver disease, diabetes with chronic complications, and renal disease was significantly higher in patients with excessive polypharmacy than in those without.

In patients without dementia, we observed higher mild disability, Medical Aid, home care, and CCI scores in patients with polypharmacy and excessive polypharmacy than those without, consistent with the findings for patients with dementia (S4 Table). Based on the CCI, higher comorbidity rates were observed in patients with polypharmacy and excessive polypharmacy than in those without (S5 Table).

The regression analysis showed that the likelihood of experiencing polypharmacy in patients with dementia increased significantly with age up to 70 ($p<0.05$), whereas it decreased at ages above 80. Those with mild disability were significantly more likely to experience polypharmacy than those without disabilities. Conversely, patients with severe disability had a significantly lower likelihood of experiencing polypharmacy. Compared to patients in the fifth NHI income quintile, Medical Aid beneficiaries with dementia were significantly more likely to experience polypharmacy [odds ratio (OR), 1.206; 95% confidence interval (CI), 1.135–1.281) and excessive polypharmacy (OR, 1.688; 95% CI, 1.590–1.792). Compared to patients who did not receive long-term care benefits, those who received home care had a significantly higher likelihood of polypharmacy and excessive polypharmacy, while those who received institutional care only had a significantly higher likelihood of experiencing polypharmacy. We found that the likelihood of experiencing polypharmacy and excessive polypharmacy increased with dementia duration and CCI score (Table 3).

**Table 1. Characteristics of dementia and non-dementia groups.**

| Characteristic | | Total | | Dementia | | Non-dementia | | P-value* |
|---|---|---|---|---|---|---|---|---|
| | | n | % | n | % | n | % | |
| Total | | 867,677 | (100.0) | 57,346 | (100.0) | 810,331 | (100.0) | |
| Polypharmacy** | 0–4 | 524,210 | (60.4) | 17,028 | (29.7) | 507,182 | (62.6) | <0.0001 |
| | 5–9 | 274,909 | (31.7) | 26,737 | (46.6) | 248,172 | (30.6) | |
| | 10+ | 68,558 | (7.9) | 13,581 | (23.7) | 54,977 | (6.8) | |
| Sex | Male | 388,789 | (44.8) | 18,344 | (32.0) | 370,445 | (45.7) | <0.0001 |
| | Female | 478,888 | (55.2) | 39,002 | (68.0) | 439,886 | (54.3) | |
| Age (mean±SD) | | 70.1 | (7.9) | 79.6 | (7.0) | 69.5 | (7.5) | <0.0001 |
| Age range | 60–64 | 274,239 | (31.6) | 2,125 | (3.7) | 272,114 | (33.6) | <0.0001 |
| | 65–69 | 189,866 | (21.9) | 3,603 | (6.3) | 186,263 | (23.0) | |
| | 70–74 | 148,178 | (17.1) | 6,619 | (11.5) | 141,559 | (17.5) | |
| | 75–79 | 123,755 | (14.3) | 13,124 | (22.9) | 110,631 | (13.7) | |
| | 80–84 | 84,352 | (9.7) | 16,724 | (29.2) | 67,628 | (8.3) | |
| | 85–89 | 39,630 | (4.6) | 12,219 | (21.3) | 27,411 | (3.4) | |
| | 90–91 | 7,657 | (0.9) | 2,932 | (5.1) | 4,725 | (0.6) | |
| Disability | Severe | 30,859 | (3.6) | 5,286 | (9.2) | 25,573 | (3.2) | <0.0001 |
| | Mild | 87,855 | (10.1) | 10,675 | (18.6) | 77,180 | (9.5) | |
| | None | 748,963 | (86.3) | 41,385 | (72.2) | 707,578 | (87.3) | |
| Region | Seoul-Metro | 385,324 | (44.4) | 21,743 | (37.9) | 363,581 | (44.9) | <0.0001 |
| | Chungcheong | 95,723 | (11.0) | 7,319 | (12.8) | 88,404 | (10.9) | |
| | Honam | 101,664 | (11.7) | 9,493 | (16.6) | 92,171 | (11.4) | |
| | Gyeongsang | 241,936 | (27.9) | 15,828 | (27.6) | 226,108 | (27.9) | |
| | Gangwon-Jeju | 43,030 | (5.0) | 2,963 | (5.2) | 40,067 | (4.9) | |
| Income level | Medical Aid | 58,010 | (6.7) | 8,452 | (14.7) | 49,558 | (6.1) | <0.0001 |
| | NHI 1st | 141,169 | (16.3) | 8,684 | (15.1) | 132,485 | (16.3) | |
| | NHI 2nd | 97,507 | (11.2) | 4,419 | (7.7) | 93,088 | (11.5) | |
| | NHI 3rd | 127,012 | (14.6) | 6,595 | (11.5) | 120,417 | (14.9) | |
| | NHI 4th | 173,556 | (20.0) | 9,348 | (16.3) | 164,208 | (20.3) | |
| | NHI 5th (richest) | 270,423 | (31.2) | 19,848 | (34.6) | 250,575 | (30.9) | |
| Type of long-term care benefits | Institutional care | 10,923 | (1.3) | 8,681 | (15.1) | 2,242 | (0.3) | <0.0001 |
| | Home care | 36,716 | (4.2) | 19,036 | (33.2) | 17,680 | (2.2) | |
| | None | 820,038 | (94.5) | 29,629 | (51.7) | 790,409 | (97.5) | |
| Duration of dementia (years) | <1 | | | 24,014 | (41.9) | | | |
| | 1–3 | | | 11,273 | (19.7) | | | |
| | 3–10 | | | 19,041 | (33.2) | | | |
| | ≥10 | | | 3,018 | (5.3) | | | |
| CCI | ≤2 | 651,670 | (75.1) | 18,898 | (33.0) | 632,772 | (78.1) | <0.0001 |
| | 3–4 | 152,305 | (17.6) | 20,579 | (35.9) | 131,726 | (16.3) | |
| | ≥5 | 63,702 | (7.3) | 17,869 | (31.2) | 45,833 | (5.7) | |

*Chi-square or *t*-test of differences between dementia and non-dementia groups

**Number of concurrently prescribed medications over 90 d

SD: standard deviation, NHI: National Health Insurance, CCI: Charlson comorbidity index

Excluding the duration of dementia, the direction of influence of age, disability, Medical Aid beneficiaries, and CCI on polypharmacy and excessive polypharmacy and excessive polypharmacy was similar between patients with and without dementia, but the magnitude of the influence was greater in patients without dementia. In contrast, patients without dementia but

**Table 2. Characteristics of patients with dementia based on polypharmacy status.**

| Characteristics | | Dementia | | Polypharmacy (5+) | | | Excessive polypharmacy (10+) | | |
|---|---|---|---|---|---|---|---|---|---|
| | | n | % | n | % | *P*-value* | n | % | *P*-value** |
| **Total** | | 57,346 | (100.0) | 40,318 | (100.0) | | 13,581 | (100.0) | |
| **Sex** | **Male** | 18,344 | (32.0) | 12,829 | (31.8) | 0.1825 | 4,622 | (34.0) | <0.0001 |
| | **Female** | 39,002 | (68.0) | 27,489 | (68.2) | | 8,959 | (66.0) | |
| **Age** | **60–64** | 2,125 | (3.7) | 1,288 | (3.2) | <0.0001 | 395 | (2.9) | <0.0001 |
| | **65–69** | 3,603 | (6.3) | 2,352 | (5.8) | | 759 | (5.6) | |
| | **70–74** | 6,619 | (11.5) | 4,649 | (11.5) | | 1,688 | (12.4) | |
| | **75–79** | 13,124 | (22.9) | 9,674 | (24.0) | | 3,560 | (26.2) | |
| | **80–84** | 16,724 | (29.2) | 12,106 | (30.0) | | 4,154 | (30.6) | |
| | **85–89** | 12,219 | (21.3) | 8,351 | (20.7) | | 2,521 | (18.6) | |
| | **90–91** | 2,932 | (5.1) | 1,898 | (4.7) | | 504 | (3.7) | |
| **Disability** | **Severe** | 5,286 | (9.2) | 3,687 | (9.1) | <0.0001 | 1,327 | (9.8) | <0.0001 |
| | **Mild** | 10,675 | (18.6) | 8,008 | (19.9) | | 3,122 | (23.0) | |
| | **None** | 41,385 | (72.2) | 28,623 | (71.0) | | 9,132 | (67.2) | |
| **Region** | **Seoul-Metro** | 21,743 | (37.9) | 15,469 | (38.4) | 0.0180 | 4,966 | (36.6) | <0.0001 |
| | **Chungcheong** | 7,319 | (12.8) | 5,101 | (12.7) | | 1,658 | (12.2) | |
| | **Honam** | 9,493 | (16.6) | 6,638 | (16.5) | | 2,556 | (18.8) | |
| | **Gyeongsang** | 15,828 | (27.6) | 11,040 | (27.4) | | 3,740 | (27.5) | |
| | **Gangwon-Jeju** | 2,963 | (5.2) | 2,070 | (5.1) | | 661 | (4.9) | |
| **Income level** | **Medical Aid** | 8,452 | (14.7) | 6,369 | (15.8) | <0.0001 | 2,824 | (20.8) | <0.0001 |
| | **NHI 1st** | 8,684 | (15.1) | 5,971 | (14.8) | | 1,950 | (14.4) | |
| | **NHI 2nd** | 4,419 | (7.7) | 2,976 | (7.4) | | 943 | (6.9) | |
| | **NHI 3rd** | 6,595 | (11.5) | 4,540 | (11.3) | | 1,434 | (10.6) | |
| | **NHI 4th** | 9,348 | (16.3) | 6,517 | (16.2) | | 2,068 | (15.2) | |
| | **NHI 5th (richest)** | 19,848 | (34.6) | 13,945 | (34.6) | | 4,362 | (32.1) | |
| **Type of long-term care benefits** | **Institutional care** | 8,681 | (15.1) | 6,427 | (15.9) | <0.0001 | 1,950 | (14.4) | <0.0001 |
| | **Home care** | 19,036 | (33.2) | 13,877 | (34.4) | | 4,960 | (36.5) | |
| | **None** | 29,629 | (51.7) | 20,014 | (49.6) | | 6,671 | (49.1) | |
| **Duration of dementia (years)** | **<1** | 24,014 | (41.9) | 16,313 | (40.5) | <0.0001 | 5,388 | (39.7) | <0.0001 |
| | **1–3** | 11,273 | (19.7) | 8,053 | (20.0) | | 2,758 | (20.3) | |
| | **3–10** | 19,041 | (33.2) | 13,744 | (34.1) | | 4,675 | (34.4) | |
| | **≥10** | 3,018 | (5.3) | 2,208 | (5.5) | | 760 | (5.6) | |
| **CCI** | **≤2** | 18,898 | (33.0) | 10,984 | (27.2) | <0.0001 | 2,085 | (15.4) | <0.0001 |
| | **3–4** | 20,579 | (35.9) | 15,021 | (37.3) | | 4,903 | (36.1) | |
| | **≥5** | 17,869 | (31.2) | 14,313 | (35.5) | | 6,593 | (48.5) | |

*Chi-square test of differences between patients with and without polypharmacy (5+)

**Chi-square test of differences between patients with and without excessive polypharmacy (10+)

NHI: National Health Insurance; CCI: Charlson comorbidity index

with severe disabilities were significantly more likely to have polypharmacy and excessive polypharmacy than those without disabilities. The likelihood of excessive polypharmacy was significantly higher in the lower NHI income groups among patients without dementia (S6 Table).

Regarding the related comorbidities, patients with dementia had a relatively higher prevalence of depression and mental health-related diseases than patients without dementia, and even among patients with dementia, the patients with polypharmacy and excessive polypharmacy had a relatively higher proportion of these comorbidities (S2, S3 and S5 Tables).

**Table 3. Factors associated with polypharmacy among patients with dementia (n = 57,346).**

| Variable | | Polypharmacy (5+) | | | Excessive polypharmacy (10+) | | |
|---|---|---|---|---|---|---|---|
| | | OR | 95% CI | *P*-value | OR | 95% CI | *P*-value |
| **Sex** | **Male (ref.)** | | | | | | |
| | **Female** | 1.052 | 1.010–1.095 | 0.0143 | 0.937 | 0.897–0.979 | 0.0034 |
| **Age** | **60–64 (ref.)** | | | | | | |
| | **65–69** | 1.189 | 1.061–1.332 | 0.0030 | 1.180 | 1.026–1.358 | 0.0208 |
| | **70–74** | 1.458 | 1.312–1.619 | <0.0001 | 1.519 | 1.337–1.726 | <0.0001 |
| | **75–79** | 1.690 | 1.530–1.866 | <0.0001 | 1.661 | 1.470–1.875 | <0.0001 |
| | **80–84** | 1.543 | 1.399–1.703 | <0.0001 | 1.483 | 1.313–1.674 | <0.0001 |
| | **85–89** | 1.216 | 1.099–1.345 | 0.0001 | 1.152 | 1.016–1.306 | 0.0269 |
| | **90–91** | 1.002 | 0.887–1.132 | 0.9724 | 0.920 | 0.789–1.072 | 0.2851 |
| **Disability** | **Severe** | 0.858 | 0.803–0.916 | <0.0001 | 0.959 | 0.893–1.030 | 0.2479 |
| | **Mild** | 1.245 | 1.184–1.309 | <0.0001 | 1.316 | 1.252–1.383 | <0.0001 |
| | **None (ref)** | | | | | | |
| **Region** | **Seoul-Metro (ref)** | | | | | | |
| | **Chungcheong** | 0.909 | 0.856–0.964 | 0.0016 | 0.966 | 0.905–1.032 | 0.3045 |
| | **Honam** | 0.884 | 0.837–0.933 | <0.0001 | 1.145 | 1.081–1.213 | <0.0001 |
| | **Gyeongsang** | 0.937 | 0.895–0.982 | 0.0060 | 1.038 | 0.987–1.092 | 0.1512 |
| | **Gangwon-Jeju** | 0.911 | 0.835–0.993 | 0.0332 | 0.928 | 0.844–1.022 | 0.1285 |
| **Income level** | **Medical Aid** | 1.206 | 1.135–1.281 | <0.0001 | 1.688 | 1.590–1.792 | <0.0001 |
| | **NHI 1st** | 0.934 | 0.882–0.988 | 0.0172 | 1.079 | 1.012–1.149 | 0.0191 |
| | **NHI 2nd** | 0.901 | 0.838–0.968 | 0.0044 | 1.015 | 0.935–1.103 | 0.7163 |
| | **NHI 3rd** | 0.953 | 0.896–1.014 | 0.1292 | 1.018 | 0.949–1.091 | 0.6229 |
| | **NHI 4th** | 0.998 | 0.944–1.055 | 0.9471 | 1.033 | 0.972–1.099 | 0.2972 |
| | **NHI 5th (richest, ref.)** | | | | | | |
| **Type of long-term care benefits** | **Institutional care** | 1.366 | 1.287–1.451 | <0.0001 | 0.986 | 0.924–1.051 | 0.6588 |
| | **Home care** | 1.266 | 1.212–1.323 | <0.0001 | 1.193 | 1.139–1.250 | <0.0001 |
| | **None (ref.)** | | | | | | |
| **Duration of dementia (years)** | **<1 (ref.)** | | | | | | |
| | **1–3** | 1.187 | 1.128–1.249 | <0.0001 | 1.194 | 1.130–1.262 | <0.0001 |
| | **3–10** | 1.216 | 1.163–1.272 | <0.0001 | 1.208 | 1.151–1.268 | <0.0001 |
| | **≥10** | 1.263 | 1.156–1.380 | <0.0001 | 1.270 | 1.157–1.393 | <0.0001 |
| **CCI** | **≤2 (ref.)** | | | | | | |
| | **3–4** | 1.960 | 1.878–2.046 | <0.0001 | 2.477 | 2.341–2.620 | <0.0001 |
| | **≥5** | 2.910 | 2.775–3.052 | <0.0001 | 4.524 | 4.280–4.782 | <0.0001 |

Ref.: reference

NHI: National Health Insurance, CCI: Charlson comorbidity index

Regarding medication composition, the proportion of antipsychotics and antidepressants was relatively high in patients with dementia than in those without (S7 Table).

## Discussion

This study examined the polypharmacy status of patients with dementia who were prescribed oral medications during outpatient treatment in 2019 and identified the associated factors using the NHIS-Senior cohort database, comprising representative data from a public single-payer in South Korea. The main findings of this study are as follows.

First, elderly patients with dementia experienced polypharmacy more than those without dementia. In patients without dementia, the proportions experiencing polypharmacy and excessive polypharmacy were 37.4% and 6.8%, respectively, while they were as high as 70.3% and 23.7%, respectively, in patients with dementia. In 2019, South Korea had a polypharmacy rate (the proportion of patients chronically prescribed five or more medications) of 70.2% for patients aged≥75 years, which was the third highest rate among OECD countries (46.7% average from 16 countries) [48]. According to a previous study that examined polypharmacy in Korean outpatients aged≥65 years using NHI claims data from 2010 to 2011, 86.4% of patients simultaneously used six or more medication, while 44.9% simultaneously used 11 or more medications [49]. Another study found that 41.8% and 14.4% of outpatients aged≥65 years were prescribed five or more and ten and more medications, respectively, for more than 90 days in 2019 [50]. These studies confirm the high prevalence of polypharmacy in the Korean elderly population. However, the present study is the first to report polypharmacy prevalence in Korean patients with dementia. Although absolute comparisons should be made with caution due to substantial age differences between patients with and without dementia, our findings indicate a higher prevalence of polypharmacy and excessive polypharmacy in patients with dementia patients. Because patients with dementia-related cognitive decline and memory loss may have more difficulty in identifying the side effects of medications compared to patients without dementia [19, 20], and polypharmacy may exacerbate the symptoms or be a risk factor for dementia [45, 51, 52], it is crucial that closer attention be paid to the risk of polypharmacy in the clinical treatment of patients dementia, in both the theoretical and practical spheres.

Second, the likelihood of polypharmacy and excessive polypharmacy increased as the duration of dementia increased even after controlling for age and CCI, which was not reported in previous studies. This suggests that more medications were prescribed for treating the behavioral and psychological symptoms of dementia the duration of dementia increased. This possibility is supported by the higher proportion of antipsychotics and antidepressants in the medication composition of polypharmacy in patients with dementia (S7 Table). The efficacy of cholinesterase inhibitors (ChEIs), a major dementia medication, is likely to diminish over time, though the long-term effects are unclear. Notable, the side effects of ChEIs increase dose-dependently, leading to increased risk of adverse drug reactions in elderly patients with moderate to severe dementia [53]. Prescription of only of memantine, a dementia medication with a different mechanism to ChEIs, has been proposed to balance treatment side effects and benefits in moderate to severe dementia [53]. Consequently, there is little support for increasing the number of medications as the duration of dementia increases. Moreover, although dementia symptoms, such as BPSD [54] may worsen as the duration of dementia increases, the suitability of long-term antipsychotic use in patients with dementia remains unclear [55]. Therefore, closer attention should be paid to the current and future composition of medications prescribed for elderly patients with dementia, especially given the vulnerability of this population.

Third, the likelihood of polypharmacy and excessive polypharmacy was especially high among Medical Aid beneficiaries, the lowest-income group, consistent with previous findings on elderly patients in South Korea [49, 56]. While previous studies compared NHI with Medical Aid beneficiaries, we further divided the NHI population into income quintiles but found no difference in the likelihood of polypharmacy and excessive polypharmacy between income quintiles in patients with dementia. The significantly higher likelihood of polypharmacy and excessive polypharmacy in Medical Aid compared to NHI beneficiaries may be due to differences in benefit systems or health status rather than solely income disparities. Because Medical Aid beneficiaries have low out-of-pocket healthcare costs, they may be able to use medical care and receive prescriptions for mediations more easily, thereby increasing the possibility of

polypharmacy. Considering that Medical Aid beneficiaries may exhibit worse health conditions, e.g., more comorbidities than the NHI population [56], greater attention should be paid to polypharmacy in this population. Despite recent efforts in managing the duplication of medication [56] and the fact that polypharmacy may be easier to manage among Medical Aid beneficiaries as they are required to use comparatively more limited medical institutions than NHI population, current policies have failed to address polypharmacy. Hence, new national policies are urgently needed to improve the management of polypharmacy in Medical Aid beneficiaries.

Fourth, both patients with and without dementia had a high likelihood of polypharmacy when receiving long-term care benefits. Compared to patients not receiving long-term care benefits, the likelihood of polypharmacy and excessive polypharmacy was significantly higher in patients receiving home care, and the likelihood of polypharmacy was significantly higher in patients receiving institutional care. Raising awareness of polypharmacy by educating long-term care providers about the potential risks could contribute to quickly recognizing and responding to the risks that may arise due to polypharmacy.

Finally, our comparison of patients with and without dementia elucidated the characteristics of polypharmacy in this cohort along with important factors that affect its prevalence in patients with dementia. Since age, disability, Medical Aid beneficiaries, and CCI had greater effects on polypharmacy in patients without dementia (S6 Table), dementia itself may be a risk factor for polypharmacy. The higher risk for polypharmacy in patients with dementia compared to those without is consistent with findings from the United Kingdom (patients aged ≥65 years registered at general practices) [8] and United States (outpatients aged ≥65 years) [21], though the cohorts in these studies had different demographic profiles and healthcare systems. Additionally, the different directions with which patient characteristics affected polypharmacy, such as excessive polypharmacy increasing with decreasing income quintiles and the presence of severe disabilities among patients without dementia, suggests that some factors may have qualitatively different effects in patients with and without dementia. This in turn suggests that differentiated approaches may be required to treat these two populations.

This study has some limitations. First, the use of a specific numeric threshold to define polypharmacy may not always be applicable, and some cases require the simultaneous use of several medications. Hence, studies on polypharmacy may benefit from distinguishing appropriate and inappropriate polypharmacy [13], which was not taken into account in this study. However, there is no clear standard for distinguishing appropriate from inappropriate polypharmacy, and given the insufficient fundamental research on the status of polypharmacy among patients with dementia in South Korea and its factors, it can be seen that research on this should be preceded. To the best of our knowledge, this study provides the first insights into the status and associated factors of polypharmacy among patients with dementia in Korea. Although many previous studies focused solely on the number of medications taken simultaneously [13], this study also considered the treatment duration (in days) which may be more appropriate for capturing clinically relevant cases. Second, polypharmacy may have been underestimated since the NHIS-Senior cohort dataset does not capture the use of over-the-counter medications and those not covered by the NHI, which neglects various potential drug interactions and adverse reactions. Nevertheless, our comparison of patients with and without dementia under the same definition of polypharmacy allowed for robust statistical analysis. Third, although we confirmed that patients with a longer duration of dementia and Medical Aid beneficiaries are more prone to polypharmacy, even when controlling for age and CCI, we did not specifically identify the key medications or clinical treatments. Fourth, the data source used in this study was sampled to represent 60–80-year-old population in 2008. Although a new sample of the population turning 60-year-old was added every year until 2019, our cohort

may not strictly represent the elderly in Korea as of 2019. However, to include elderly of over 80 years and analyze the most recent situation, we analyzed data from 2019. Even though we attempted to include the maximum possible age within the data, the oldest individual in 2019 was 91; therefore, our study findings do not represent elderly individuals aged≥92 years. This along with the fact that we only considered Korean patients, limits the generalization of our results. Fifth, we did not include the results of the non-dementia group in the main table while analyzing the factors associated with polypharmacy. A comprehensive comparison could further elucidate the risk factors for polypharmacy in patients with dementia. However, considering that dementia duration, only applied to patients with dementia, we chose to present the results for patients with and without dementia separately.

Despite these limitations, we used the same framework for study polypharmacy in patients with and without dementia. Our comparison between patients with and without dementia suggested that polypharmacy should be approached differently between these patient populations, barring further qualitative research on the individual appropriateness of polypharmacy. Further research is also needed on the long-term consequences of polypharmacy in dementia patients.

## Conclusions

With the increasing global incidence of dementia, greater attention should be paid to polypharmacy among affected patients from both clinical and economic perspectives. However, studies on polypharmacy in patients with dementia have been insufficient. To address this gap in knowledge, we examined the polypharmacy status and its associated factors in dementia in the NHIS-Senior cohort based on the NHI claims data in South Korea. Our findings emphasize the need for closer scrutiny in the simultaneous prescription of dementia and related medications, and provide fundamental evidence in this regard.

## Supporting information

**S1 Table. Codes used to define patients with dementia.**
(DOCX)

**S2 Table. Distribution of comorbidities among study population.**
(DOCX)

**S3 Table. Distribution of comorbidities among patients with dementia based on polypharmacy status.**
(DOCX)

**S4 Table. Characteristics of patients without dementia based on polypharmacy status.**
(DOCX)

**S5 Table. Distribution of comorbidities among patients without dementia based on polypharmacy status.**
(DOCX)

**S6 Table. Factors associated with polypharmacy: Comparison between patients with and without dementia.**
(DOCX)

**S7 Table. Distribution of medications included in polypharmacy.**
(DOCX)

## Acknowledgments

This study used NHIS-Senior data (NHIS-2022-2-352) provided by the National Health Insurance Service (NHIS) of South Korea.

## Author Contributions

**Conceptualization:** Hye-Jae Lee.

**Data curation:** Hea-Lim Kim.

**Formal analysis:** Hea-Lim Kim.

**Funding acquisition:** Hye-Jae Lee.

**Investigation:** Hye-Jae Lee.

**Methodology:** Hea-Lim Kim, Hye-Jae Lee.

**Project administration:** Hea-Lim Kim.

**Supervision:** Hye-Jae Lee.

**Visualization:** Hea-Lim Kim.

**Writing – original draft:** Hea-Lim Kim.

**Writing – review & editing:** Hea-Lim Kim, Hye-Jae Lee.

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
