## [Decision Letter · Decision Letter 0]

12 Dec 2023

PONE-D-23-24901Polypharmacy patterns and associated factors in South Korean elderly patients with dementia: An analysis using National Health Insurance claims dataPLOS ONE

Dear Dr. Lee,

Thank you for submitting your manuscript to PLOS ONE. After careful consideration, we feel that it has merit but does not fully meet PLOS ONE’s publication criteria as it currently stands. Therefore, we invite you to submit a revised version of the manuscript that addresses the points raised during the review process.

Please find the reviewers' comments below and submit a point-by-point response.

We look forward to receiving your revised manuscript.

Kind regards,

Muhammad Eid Akkawi

Academic Editor

PLOS ONE

3. PLOS requires an ORCID iD for the corresponding author in Editorial Manager on papers submitted after December 6th, 2016. Please ensure that you have an ORCID iD and that it is validated in Editorial Manager. To do this, go to ‘Update my Information’ (in the upper left-hand corner of the main menu), and click on the Fetch/Validate link next to the ORCID field. This will take you to the ORCID site and allow you to create a new iD or authenticate a pre-existing iD in Editorial Manager. Please see the following video for instructions on linking an ORCID iD to your Editorial Manager account: " ext-link-type="uri" xlink:type="simple">https://www.youtube.com/watch?v=_xcclfuvtxQ".

4. We are unable to open your Supporting Information file [Figure 1.pptx]. Please kindly revise as necessary and re-upload.

Additional Editor Comments:

- The use of the term "polypharmacy patterns" is not tally with the results and discussion. No results were reported about the types of medications or dosing regimens. Therefore, please remove the word patterns and stick to "polypharmacy" throughout the manuscript.

- L150: to stay consistent, please report the percentages for polypharmacy instead of non-polypharmacy.

- Do not repeat the results in the text and tables. Just highlight the important findings in the text or findings not stated in the tables.

- Table 1: Add a comparison of the mean age between the two groups.

- Include the number of patients in the table captions.

- Make the abstract structured with conclusion.

- English language proofreading is needed.

Reviewers' comments:

Reviewer's Responses to Questions

**Comments to the Author**

1. Is the manuscript technically sound, and do the data support the conclusions?

Reviewer #1: Partly

Reviewer #2: Yes

2. Has the statistical analysis been performed appropriately and rigorously? 

Reviewer #1: Yes

Reviewer #2: Yes

3. Have the authors made all data underlying the findings in their manuscript fully available?

Reviewer #1: No

Reviewer #2: Yes

4. Is the manuscript presented in an intelligible fashion and written in standard English?

Reviewer #1: No

Reviewer #2: Yes

5. Review Comments to the Author

Reviewer #1: I thank the author for this interesting work about polypharmacy and dementia. This work is relevant as it found consistent result with literature with over-prescription in patients suffering from dementia. A major strength of this study is that it was performed on a large sample of a national cohort. However, a better explanation of the method used, and an update of the organization of the results and discussion can improve this study quality and it's understanding.

Major comments:

1. The goal announced is “to discern patterns and associated factors of polypharmacy among outpatient suffering from dementia “. Results and discussion sections show first a comparison between dementia vs non-dementia patients then polypharmacy vs non-polypharmacy in patients suffering from dementia. Both groups can be compared to achieve the announced goal, but they answer different questions. Thus, the author should either improve the formulation of the goal/question and reorganize the methods, results, and discussion sections accordingly.

2. In line with the previous comment, the limit discussed l. 274 might be the most significant. Yet, it is discussed as the last one and the justification provided is not satisfactory. It would be possible either to attribute the value 0 for “Duration of dementia” to negate the variable for non-demented patients or to treat missing values for this variable as a level (if there are no missing values in the dementia patients’ group).

Minor comments

1. The manuscript should be reviewed by an English native speaker to improve its general understanding and quality.

2. The author should specify the ethic committee and its decision date and decision reference

3. The author used the codes F00, F01, F02, F03, G30 and G31.82, which refer mainly to Alzheimer’s disease However, these codes do not include cognitive disorder or impairment, which is a major selection bias. Is it possible for the author to include those missing patients? If not, could they provide justification and/or specify in the method section that these patients were not included?

4. The author defined disability using the disability grades used in South Korea. Can they provide how these grades are assessed?

5. Can the author precise the ranking of the quantiles used for the income level? It is only precise in the results section but a phrase such as (“the fifth quantile represents the highest income”) could be added in the method section.

6. In the table 1, the term Institutional care was used whereas “facility service” is used in the method. The authors should stay consistent.

7. Missing values assessment, analysis and consideration should be provided in the method section.

8. Can the author precise how the CCI variable was dichotomized? The result section shows 3 categories, but it was not specified why/how these cut-offs were chosen.

9. The models used may include redundant variables (CCI and dementia, disability, and dwelling accommodation, …). Have the authors assessed multicollinearity? If so, can they specify it in the method section?

10. The authors should provide in supplementary material comorbidities’ details (overall and groups/sub-groups)

11. In the discussion, the authors assumed that the difference of polypharmacy in dementia patients may be because of the prescription of BPSD controlling drugs. As the author have access to prescriptions records, it will strengthen authors’ conclusion if drug classes comparison was provided.

12. In the discussion, the authors discussed the issue of income when comparing Medical Aid vs NHI population. However, the analysis showed an OR 1 for NIH 1 to 3 compared when compared to NIH 5. Could this author add a section to discuss this result as well?

13. This author should give practical implication of their findings.

Reviewer #2: The article you've presented appears to be a comprehensive and methodologically sound study focusing on polypharmacy among elderly patients with dementia in South Korea. Here are some reviewer comments that could be helpful:

Strengths:

Relevance and Timeliness: The topic of polypharmacy in dementia patients is both timely and relevant, especially considering the aging population and the increase in dementia cases globally.

Data Source and Population: The use of the National Health Insurance Service-Senior cohort database provides a robust and representative sample for this type of research.

Methodological Rigor: The study's methodology, including the definition of polypharmacy and excessive polypharmacy, patient categorization, and statistical analysis, seems thorough and well-justified.

Policy Implications: The findings of this study have significant implications for healthcare policy and practice, particularly in countries with rapidly aging populations.

Areas for Improvement:

Clarity in Introduction: The introduction could benefit from a clearer articulation of the research gap. While it mentions the scarcity of studies in this area, a more explicit statement of what specifically this study adds would be helpful.

Broader Contextualization: The discussion could explore how these findings relate to or differ from studies in other countries, especially those with different healthcare systems or demographic profiles.

Addressing Limitations: While some limitations are implicit (such as the study's focus on South Korea), it would be beneficial for the authors to explicitly state these limitations and discuss how they might affect the generalizability of the findings.

Potential Biases: Any potential biases in the data source or methodology should be acknowledged and discussed, such as selection bias in the cohort or limitations in the diagnostic codes used.

Implications for Future Research: The article could offer more detailed suggestions for future research, such as exploring interventions to reduce polypharmacy risks or examining the impact of polypharmacy on specific outcomes in dementia patients.

Editorial Suggestions:

Abstract Structure: The abstract could be structured more clearly with distinct sections for background, methods, results, and conclusions to enhance readability.

Consistency in Terminology: Ensure consistency in the use of terms and definitions throughout the paper.

References: Check for the latest references, especially if there have been significant developments in the field since the original literature review was conducted.

Overall, the article contributes valuable insights into an under-researched area and has the potential to inform clinical practices and policy decisions in the care of dementia patients.

6. PLOS authors have the option to publish the peer review history of their article (what does this mean?). If published, this will include your full peer review and any attached files.

Reviewer #1: **Yes: **Edouard Baudouin

Reviewer #2: **Yes: **Ali Haider Mohammed

---

## [Author Response · Author response to Decision Letter 0]

24 Feb 2024

Response to Comments regarding Journal Requirements

● Comment 1. 

Thank you for your guidance. We revised the text to meet the style requirements of PLOS ONE, as indicated in the template link provided. To avoid confusing the reviewers with the numerous tracked changes associated with content and style changes, the formatting changes were made with Track Changes Off. Formatting was performed according to the templates provided by PLOS ONE. Regarding the reference citation format, parentheses were changed to square brackets, but these changes were not separately indicated due to the use of bibliographic software. 

● Comment 2. 

In your Data Availability statement, you have not specified where the minimal data set underlying the results described in your manuscript can be found. PLOS defines a study's minimal data set as the underlying data used to reach the conclusions drawn in the manuscript and any additional data required to replicate the reported study findings in their entirety. All PLOS journals require that the minimal data set be made fully available. […] If there are ethical or legal restrictions to sharing your data publicly, please explain these restrictions in detail. Please see our guidelines for more information on what we consider unacceptable restrictions to publicly sharing data.

There are legal restrictions on publicly sharing the data used in this study. Therefore, referring to the guide provided, we supplemented the data availability statement as follows, focusing on the reasons why the data cannot be shared and how other researchers can access the data. 

Revised Data Availability statement: 

“Data availability: This study used the NHIS-Senior cohort database curated by the National Health Insurance Service (NHIS) of Korea. This data cannot be shared publicly due to NHIS regulations. We obtained approval for use of the data after review by the Institutional Review Board of the affiliated institution and a separate review by the NHIS, and paid an access fee. We were able to access the data for the approved period through the remote service operated by the NHIS. Those who are eligible for use of the National Health Information Data, as stipulated by the NHIS, can access the NHIS-Senior cohort database following the same procedure mentioned here. Applications can be made with the National Health Insurance Sharing Service (https://nhiss.nhis.or.kr/bd/ab/bdaba021eng.do).”

● Comment 3. 

PLOS requires an ORCID iD for the corresponding author in Editorial Manager on papers submitted after December 6th, 2016. Please ensure that you have an ORCID iD and that it is validated in Editorial Manager.

Thank you for this guidance. We have resolved the issue of linking the ORCID of the corresponding author (Hye-Jae Lee, 0000-0002-1010-4925) with the Editorial Office's assistance. It has been confirmed that the ORCID is properly linked with the submission ID (hjlee1) in Editorial Manager.

● Comment 4.

We are unable to open your Supporting Information file [Figure 1.pptx]. Please kindly revise as necessary and re-upload.

 We apologize for the inconvenience. We saved and reuploaded Figure 1 in TIFF format. 

Response to Comments from the Editor

● Comment 1.

The use of the term “polypharmacy patterns” is not tally with the results and discussion. No results were reported about the types of medications or dosing regimens. Therefore, please remove the word patterns and stick to “polypharmacy” throughout the manuscript. 

Thank you for your reasonable comment. As suggested, we revised the text to avoid the expression “polypharmacy patterns” and used the expression “polypharmacy” throughout the text. The title was also revised to delete the word “patterns.”

Revised title:

“Polypharmacy and associated factors in South Korean elderly patients with dementia: An analysis using National Health Insurance claims data”

Revised text on page 2, line 22 – 23: 

“This study aimed to characterize polypharmacy and associated factors among elderly patients with dementia in South Korea…”

Revised text on page 2, line 29 – 30:

“We compared the prevalence of polypharmacy between patients with and without dementia and identified the associated factors using a logistic regression model.” 

Revised text on page 4 – 5, line 67 – 69: 

“Previous research has primarily focused on nursing home settings or potentially inappropriate medications, while studies on polypharmacy and associated factors in outpatient patients with dementia are relatively scarce.”

Revised text on page 4, line 78 – 81:

“The study of polypharmacy in patients with dementia could provide valuable information not only for clinical practitioners and policy makers in Korea, where the population is aging rapidly, but also in other countries with slower population aging rates.”

Revised text on page 4, line 85 – 88:

“To bridging this gap in knowledge, we aimed to determine the status and associated factors of polypharmacy among outpatients with dementia in 2019, using data sourced from the National Health Insurance Service (NHIS)-Senior cohort database.”

Revised text on page 16, line 268 – 271:

“This study examined the polypharmacy status of patients with dementia who were prescribed oral medications during outpatient treatment in 2019 and identified the associated factors using the NHIS-Senior cohort database, comprising representative data from a public single-payer in South Korea.”

● Comment 2.

L150: to stay consistent, please report the percentages for polypharmacy instead of non-polypharmacy.

The sentence was revised as follows, with additional information provided for clarity. 

Revised text on page 10, line 207 – 209:

“Polypharmacy was recorded in 70.3% and 37.4% of patients with and without dementia, respectively, whereas excessive polypharmacy was recorded in 23.7% and 6.8% of patients with and without dementia, respectively (Table 1).” 

● Comment 3. 

Do not repeat the results in the text and tables. Just highlight the important findings in the text or findings not stated in the tables.

We agree that the original text may have repeated some results from the tables, and have revised the descriptions of the tables to focus on important findings not stated in the tables.

Revised text on page 10, line 199 – 209:

“The patients with and without dementia showed significant differences in several characteristics (Table 1). Specifically, the dementia group was, on average, 10 years older than the non-dementia group. Additionally, 10.0% of the dementia group fell into the 60s age range, compared to 56.6% of the non-dementia group. The proportion of individuals with disabilities in the dementia group was 27.8%, which was more than twice that in the non-dementia group (12.7%). Long-term care benefits were provided to 48.3% of patients with dementia, but only 2.5% of patients without dementia. Nearly 33.0% and 78.1% of patients with and without dementia had a CCI score ≤2, respectively. (The detailed distribution of comorbidities included in the CCI is presented in S2 Table.) Polypharmacy was recorded in 70.3% and 37.4% of patients with and without dementia, respectively, whereas excessive polypharmacy was recorded in 23.7% and 6.8% of patients with and without dementia, respectively (Table 1).” 

Revised text on page 11, line 214 – 225:

“There was no significant difference in the distribution of sexes between patients with and without polypharmacy among dementia group (Table 2). However, the proportion of males was significantly higher in patients with excessive polypharmacy than in those without excessive polypharmacy (34.0% compared to 31.4%). Mild disability was observed among 23.0% of patients with excessive polypharmacy, which was significantly higher than that among patients without excessive polypharmacy (17.3%). Medical Aid beneficiaries were 20.8% of patients with excessive polypharmacy (compared to 12.9% of patients without excessive polypharmacy) and 15.8% of patients with polypharmacy (compared to 12.2% of patients without polypharmacy). In terms of long-term care benefits, home care benefits were provided to 36.5% and 32.2% of patients with and without excessive polypharmacy, respectively. Patients with polypharmacy and excessive polypharmacy showed a high CCI. A CCI score≥5 was observed in 48.5% and 25.8% of patients with and without excessive polypharmacy, respectively (Table 2).” 

Revised text on page 14, line 238 – 249:

“The regression analysis showed that the likelihood of experiencing polypharmacy in patients with dementia increased significantly with age up to 70 (p0.05), whereas it decreased at ages above 80. Those with mild disability were significantly more likely to experience polypharmacy than those without disabilities. Conversely, patients with severe disability had a significantly lower likelihood of experiencing polypharmacy. Compared to patients in the fifth NHI income quintile, Medical Aid beneficiaries with dementia were significantly more likely to experience polypharmacy [odds ratio (OR), 1.206; 95% confidence interval (CI), 1.135–1.281) and excessive polypharmacy (OR, 1.688; 95% CI, 1.590–1.792). Compared to patients who did not receive long-term care benefits, those who received home care had a significantly higher likelihood of polypharmacy and excessive polypharmacy, while those who received institutional care only had a significantly higher likelihood of experiencing polypharmacy. We found that the likelihood of experiencing polypharmacy and excessive polypharmacy increased with dementia duration and CCI score (Table 3).”

● Comment 4. 

Table 1: Add a comparison of the mean age between the two groups.

We added the mean age to Table 1, and also added a description in the text. We considered adding the mean age to Table 2 but, because the difference in mean age was not substantial, we avoided this revision to keep the table from exceeding one page. For reference, the average age (standard deviation) of each group was 79.6 years (7.0) in the dementia group, 79.6 years (6.8) in dementia patients with polypharmacy, and 79.3 years (6.5) in dementia patients with excessive polypharmacy. 

● Comment 5.

Include the number of patients in the table captions.

We added the number of patients to the caption of Table 3. We did not add it to Tables 1 and 2 because the number of patients in each group is included inside the table. 

● Comment 6.

Make the abstract structured with conclusion.

We structured the abstract by adding subheadings and separated the conclusion.

Revised Abstract on page 2 – 3, line 17 – 41: 

The revised abstract is not cited here.

● Comment 7. 

English language proofreading is needed.

The manuscript was revised by an English language professional. We attached the certificate of English language proofreading.

Response to the Comments from Reviewer 1

Major comments: 

● Comment 1. 

The goal announced is “to discern patterns and associated factors of polypharmacy among outpatient suffering from dementia”. Results and discussion sections show first a comparison between dementia vs non-dementia patients then polypharmacy vs non-polypharmacy in patients suffering from dementia. Both groups can be compared to achieve the announced goal, but they answer different questions. Thus, the author should either improve the formulation of the goal/question and reorganize the methods, results, and discussion sections accordingly.

Thank you for your important advice. The main objective of this study was to elucidate the characteristics of polypharmacy in patients with dementia. Accordingly, we compared the characteristics and prevalence of polypharmacy between patients with and without dementia to identify key factors, such as age and comorbidities. The purpose of the study was revised to clearly highlight this aspect, and the Methods, Results, and Discussion sections were supplemented for clarity. 

Revised text on page 5, line 85 – 89: 

“To bridging this gap in knowledge, we aimed to determine the status and associated factors of polypharmacy among outpatients with dementia in 2019, using data sourced from the National Health Insurance Service (NHIS)-Senior cohort database. To elucidate the characteristics, prevalence, and risk factors of polypharmacy in dementia, we compared the data between patients with and without dementia.”

Revised text on page 8, line 171 – 174:

“For comparison, the second and third analyses were also conducted in patients without dementia; however, considering the difficulty in defining the duration of dementia in these patients, the analyses excluded this variable, and again, for comparison, the third analysis was also conducted excluding this variable in patients with dementia.”

Revised text on page 14, line 233 – 235:

“In patients without dementia, we observed higher mild disability, Medical Aid, home care, and CCI scores in patients with polypharmacy and excessive polypharmacy than those without, consistent with the findings for patients with dementia (S4 Table).” 

Revised text on page 15, line 253 – 259: 

“Excluding the duration of dementia, the direction of influence of age, disability, Medical Aid beneficiaries, and CCI on polypharmacy and excessive polypharmacy and excessive polypharmacy was similar between patients with and without dementia, but the magnitude of the influence was greater in patients without dementia. In contrast, patients without dementia but with severe disabilities were significantly more likely to have polypharmacy and excessive polypharmacy than those without disabilities. The likelihood of excessive polypharmacy was significantly higher in the lower NHI income groups among patients without dementia (S6 Table).”

Revised text on page 18, line 332 – 344:

“Finally, our comparison of patients with and without dementia elucidated the characteristics of polypharmacy in this cohort along with important factors that affect its prevalence in patients with dementia. Since age, disability, Medical Aid beneficiaries, and CCI had greater effects on polypharmacy in patients without dementia (S6 Table), dementia itself may be a risk factor for polypharmacy. The higher risk for polypharmacy in patients with dementia compared to those without is consistent with findings from the United Kingdom (patients aged ≥65 years registered at general practices) [8] and United States (outpatients aged ≥65 years) [21], though the cohorts in these studies had different demographic profiles and healthcare systems. Additionally, the different directions with which patient characteristics affected polypharmacy, such as excessive polypharmacy increasing with decreasing income quintiles and the presence of severe disabilities among patients without dementia, suggests that some factors may have qualitatively different effects in patients with and without dementia. This in turn suggests that differentiated approaches may be required to treat these two populations.” 

● Comment 2. 

In line with the previous comment, the limit discussed l. 274 might be the most significant. Yet, it is discussed as the last one and the justification provided is not satisfactory. It would be possible either to attribute the value 0 for “Duration of dementia” to negate the variable for non-demented patients or to treat missing values for this variable as a level (if there are no missing values in the dementia patients’ group).

Thank you for this important advice. We added the results for the logistic regression analysis on the factors influencing polypharmacy in patients without dementia as supplementary material. Since the duration of dementia was ambiguous in these patients, the analysis was conducted without the duration of dementia; for comparison, we also presented the analysis results without this variable for patients with dementia

---

## [Decision Letter · Decision Letter 1]

2 Apr 2024

Polypharmacy and associated factors in South Korean elderly patients with dementia: An analysis using National Health Insurance claims data

PONE-D-23-24901R1

Dear Dr. Lee,

We’re pleased to inform you that your manuscript has been judged scientifically suitable for publication and will be formally accepted for publication once it meets all outstanding technical requirements.

Kind regards,

Muhammad Eid Akkawi

Academic Editor

PLOS ONE

Additional Editor Comments:

In the future, please consider avoiding use of the term "elderly" as it may have negative connotation.

Reviewers' comments:

Reviewer's Responses to Questions

**Comments to the Author**

1. If the authors have adequately addressed your comments raised in a previous round of review and you feel that this manuscript is now acceptable for publication, you may indicate that here to bypass the “Comments to the Author” section, enter your conflict of interest statement in the “Confidential to Editor” section, and submit your "Accept" recommendation.

Reviewer #1: All comments have been addressed

2. Is the manuscript technically sound, and do the data support the conclusions?

Reviewer #1: Yes

3. Has the statistical analysis been performed appropriately and rigorously? 

Reviewer #1: Yes

4. Have the authors made all data underlying the findings in their manuscript fully available?

Reviewer #1: Yes

5. Is the manuscript presented in an intelligible fashion and written in standard English?

Reviewer #1: Yes

6. Review Comments to the Author

Reviewer #1: Thank you for your detailed answers and corrections. However, authors should be advised that the word "Elderly" can have a negative connation and should avoid its use in future papers

7. PLOS authors have the option to publish the peer review history of their article (what does this mean?). If published, this will include your full peer review and any attached files.

Reviewer #1: **Yes: **Edouard Baudouin
